# Identification of Key Genes Related to the Prognosis of Esophageal Squamous Cell Carcinoma Based on Chip Re-Annotation

**Meiqi Wang** [1,2,3,†], **Dan Liu** [3,†], **Yunchuanxiang Huang** [3,4] , **Ziyi Jiang** [3], **Feng Wu** [3,5], **Yu Cen** [3]
**and Lan Ma** [1,2,3,4,5,6,*]

1　Department of Chemistry, Tsinghua University, Beijing 100084, China; wmq18@mails.tsinghua.edu.cn
2　State Key Laboratory of Chemical Oncogenomics, Tsinghua Shenzhen International Graduate School, Tsinghua University, Shenzhen 518055, China
3　Institute of Biopharmaceutical and Health Engineering, Tsinghua Shenzhen International Graduate School, Tsinghua University, Shenzhen 518055, China; danieliu091@aliyun.com (D.L.); hycx19@mails.tsinghua.edu.cn (Y.H.); jiang-zy20@mails.tsinghua.edu.cn (Z.J.); wf19@mails.tsinghua.edu.cn (F.W.); cen.yu@sz.tsinghua.edu.cn (Y.C.)
4　Tsinghua-Berkeley Shenzhen Institute, Tsinghua University, Shenzhen 518055, China
5　School of Life Science, Tsinghua University, Beijing 100084, China
6　Institute of Biomedical Health Technology and Engineering, Shenzhen Bay Laboratory, Shenzhen 518132, China
*　Correspondence: malan@sz.tsinghua.edu.cn; Tel.: +86-755-26033033
†　Meiqi Wang and Dan Liu are co-first authors.

**Abstract:** Esophageal cancer (EC) is one of the deadliest cancers worldwide. However, reliable biomarkers for early diagnosis, or those for the prognosis of therapy, remain unfulfilled goals for its subtype esophageal squamous cell carcinoma (ESCC). The purpose of this study was to identify reliable biomarkers for the diagnosis and prognosis of ESCC by gene chip re-annotation technique and downstream bioinformatics analysis. In our research, the GSE53624 dataset was downloaded from the GEO database. Then, we reannotated the gene expression probe and obtained the gene expression matrix. Differential expressed genes (DEGs) were found by R packages and they were subjected to Gene Ontology enrichment analysis and protein–protein interaction (PPI) network construction. As a result, a total of 28,885 mRNA probes were reannotated, among which 210 down-regulated and 80 up-regulated DEGs were screened out. By combining these genes set in clinical prognosis information and Western blot analysis, we found four genes with diagnostic and prognostic significance, including MMP13, SPP1, MMP10, and COL1A1. Furthermore, markers of infiltrating immune cells exhibited different DEG-related immune infiltration patterns.

**Keywords:** esophageal squamous cell carcinoma; chip re-annotation; DEGs; bioinformatics analyses; prognostic biomarkers

## 1. Introduction

Esophageal cancer (EC) is the seventh most common cancer globally, with an estimated 572,000 cases and 509,000 deaths in 2018 [1]. According to the current reports [1–3], the mortality rate of esophageal cancer patients is in the front row of malignant tumors, seriously endangering people's quality of life. A newly released analysis of China's malignant cancer epidemic has esophageal cancer as the sixth most common cancer in China, with 250,000 patients dying of esophageal cancer every year [4]. Unlike Western countries, China has more EC patients with esophageal squamous cell carcinoma (ESCC), with the proportion reaching 90% [2,5]. The exploration of prognostic markers for tumor diagnosis in ESCC patients can improve the diagnosis and treatment effect of EC.

In recent years, represented by gene chip and the next-generation sequencing of rapidly developing high throughput sequencing technology, and used to analyze the high-

throughput data processing of bioinformatics research for cancer basic and clinical research provides a new train of thought and reference [6]. While esophageal adenocarcinoma (EAC) is the dominant subtype of esophageal cancer in developed countries [7], the reference of the significance for esophageal squamous carcinoma research was extremely limited. The study on ESCC lacks effective high-throughput sequencing data. Hence, the research on prognostic markers of ESCC is of great clinical significance. In this work, we re-annotated the gene chip used in the database to detect the LncRNA of ESCC by using bioinformatics technology, successfully obtained the mRNA expression profile in the study sample. We also performed a bioinformatics analysis and prognosis analysis on the expression profile based on the clinical data provided in the database, providing the value data for further research and clinical reference.

## 2. Materials and Methods

### 2.1. Gene Expression Data Set

The GSE53624 [8] Gene Expression profile data set of patients with ESCC was downloaded from the Gene Expression Omnibus (GEO) database (https://www.ncbi.nlm.nih.gov/geo/query/acc.cgi?acc=GSE53624 (accessed on 4 December 2020)). This data set was profiled using the Agilent-038314 CBC Homo sapiens V2.0 platform (GPL18109; Agilent Technologies, Inc., Santa Clara, CA, USA). There were 119 patients with ESCC included in this data set. The clinical characteristics of the ESCC patients in this study are shown in Table S1.

We downloaded the CEL file containing gene expression information and the RMA method was used for the background correction and standardization to obtain the gene expression matrix containing all the samples. The corresponding excel file containing clinical prognosis information was subsequently downloaded.

### 2.2. Chip Re-Annotation

In addition to probes for non-coding genes, Agilent has also designed a large number of detectable probes for the expression of unknown sequence tags (EST) in the human genome, most of which have been proven to be complementary gene sequences. In this study, we re-annotated the mRNA of these probes, with the specific process as follows:

1. Downloaded the matrix file of these unknown expression sequence tags and obtained the nucleic acid sequence of these probes;
2. SeqMap software [9] was used to match the nucleic acid sequences of these probes to the human genome library (ENCODE database, version 31, CA, USA) [10,11] which required matching sequences and no mismatches, and obtained the corresponding chromosome positions of the probes;
3. Removed probes matching lncRNA at the same time. A total of 28,885 reannotated mRNA probes were obtained, none of which duplicated each other.

### 2.3. Differential Expression Analysis and Functional Enrichment Analysis

The R package "limma" [12] was used to normalize, log transform, and analyze the differential expression of the original array data. The *p*-value and log2 fold change (log2FC) were calculated. Subsequently, |log2FC| > 3 and $p < 0.05$ were considered to be differentially expressed. R packages "clusterProfiler" [13], "ggplot2" [14], and "DOSE" [15] were used to perform the Gene Ontology (GO) enrichment analysis and Kyoto Encyclopedia of Genes and Genome (KEGG) pathway enrichment analysis on the DEGs. Only GO terms and KEGG pathways with $p < 0.05$ were considered statistically significant and selected for visualization.

### 2.4. Construction of Protein-Protein Interaction Network (PPI)

To further clarify the underlying molecular mechanism and gene effect of ESCC, we used the STRING database (http://www.string-db.org/ (accessed on 3 January 2021)) to retrieve the DEGs, as mentioned above, and construct a PPI network of DEGs [16].

The obtained PPI network was visualized using Cytoscape software (version 3.7.0, Free Software Foundation, Inc. Boston, MA, USA) [17]. Then, we analyzed the network's topological properties using the cytoHubba plugin (Free Software Foundation, Inc. Boston, MA, USA), calculated and sequenced the node scores of each gene, and visualized the interactions among the top ten key genes [18].

### 2.5. Identification of Prognostic Genes in Key Genes

To further obtain the genes related to prognosis in DEGs, we extracted the patient prognosis information corresponding to each chip in the data set. Perl language was used to extract the expression level of the DEGs in each sample in the gene expression array and the patient prognosis information provided in each sample was linked at the same time. R packages "survival" [19] and "ggplot2" [14] were used to conduct a survival curve analysis and log-rank tests for the above corresponding data. The significant cut-off was taken and $p < 0.05$.

### 2.6. Exploration of the Diagnostic Efficacy of Prognostic Genes

Subsequently, to explore the diagnostic efficacy of the prognostic genes, we conducted receiver operating characteristic (ROC) curves based on the specific gene expression in all gene chips. Area Under the Curve (AUC) was used to assess the diagnostic efficacy of the genes. AUC > 0.9 indicated a high diagnostic efficiency.

### 2.7. Cell Culture

Human esophageal carcinoma cell line, ESCC cell lines (KYSE-150 and KYSE-510), and normal esophageal cell line (Het-1A) were obtained from ATCC (Manassas, VA, USA). ESCC cell lines were cultured in RPMI 1640 Medium (Gibco, GlutaMAX™, #61870036, Waltham, MA, USA) with 10% fetal bovine serum (Gibco, #10099141, Waltham, MA, USA) and 1% antibiotics (penicillin and streptomycin, Gibco, #15070063, Waltham, MA, USA). Het-1A cell line was cultured in Dulbecco's Modified Eagle's Medium (DMEM, Gibco, High Glucose, GlutaMAX™, #10566016, Waltham, MA, USA) with 10% FBS and 1% antibiotics. All cells were maintained in a humidified atmosphere containing 5% $CO_2$ at 37 °C.

### 2.8. Validation Using Independent External Database and Western Blot

The online database Gene Expression Profiling Interactive Analysis 2 (GEPIA2) (http://gepia2.cancer-pku.cn/ (accessed on 10 February 2021)) was used to analyze the differential expression of ELN, MMP13, SPP1, MMP10, COL1A1, CSF2, MMP1, SERPINE1, CXCL8, MMP3.

Western blot analysis was also performed using the standard procedures. KYSE-150, KYSE-510, and Het-1A cells were lysed with RIPA buffer (Solarbio, #R0010, Beijing, China). Protein was extracted from the cells and the concentration was determined using the BCA Protein Assay Kit (Solarbio, #PC0020, Beijing, China). Protein was separated using 12% SDS-PAGE and transferred onto 0.45 μm PVDF membranes (Millipore, #IPVH00010, Burlington, MA, USA). The membranes were subsequently blocked with 5% fat-free milk (Sangon, #A600669, Shanghai, China) for 1.5 h at room temperature (RT) and incubated at 4 °C overnight with the following primary antibodies: Anti-MMP13 (1:1000; Proteintech, #18165-1-AP, Rosemont, IL, USA), Anti-SPP1 (1:1000; Proteintech, #22952-1-AP, Rosemont, IL, USA), Anti-COL1A1 (1:1000; Proteintech, #67288-1-Ig, Rosemont, IL, USA), and Anti-MMP10 (1:1000; Bioss, #bs-1344R, Beijing, China). The next day, following washing 10 times with 1x TBST (Solarbio, #T1085, Beijing, China), the membranes were incubated with secondary antibodies (1:2500; ZSGB-BIO, #ZB-5301, #ZB-5305, Beijing, China) for 1.5 h at RT. Immunoreactive bands were visualized using Immobilon Western Chemiluminescent HRP Substrate (Millipore, #WBKLS0500, Burlington, MA, USA).

## 3. Results

### 3.1. LncRNA Probe Reannotation

The base sequences of all probes were compared with the human genome library using SeqMap software, and lncRNA-related probes were excluded. A total of 28,885 probe sites containing different genetic information were re-annotated (Figure 1).

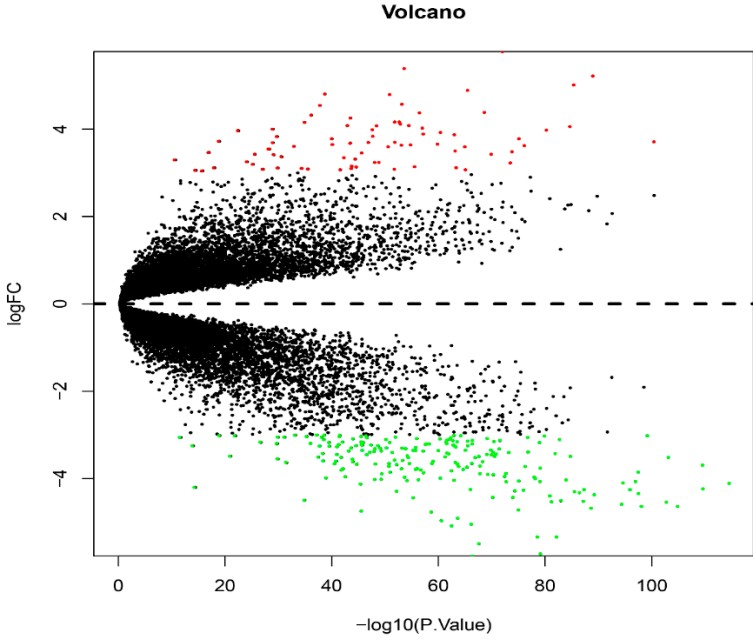

**Figure 1.** The Volcano plot of differential genes. 210 down-regulated genes and 80 up-regulated differential genes were screened out ($|\log2FC| > 3$, $p < 0.05$).

### 3.2. The Acquisition of Differential Genes

A total of 210 down-regulated and 80 up-regulated DEGs were obtained by R package "limma". Subsequently, a heatmap containing the DEGs in each sample was drawn based on the variation multiples of the DEGs in the cancer samples and the normal samples. The cluster analysis and heatmap showed that these gene changes could distinguish between cancer samples and normal samples (Figure 2).

### 3.3. Genetic Ontology (GO) Analysis

Next, we focused on the potential function of these DEGs. To clarify this issue, we performed GO analysis on these DEGs and visualized the functional clusters enriched in biological process (BP), cell composition (CC), and molecular function (MF) ($p < 0.05$). The results showed that these DEGs were enriched in many different functional clusters. Specifically, in the BP term, the DEGs were concentrated in the extracellular matrix organization, extracellular structural organization, retinoic acid metabolic process, collagen metabolic process, extracellular matrix disassembly, etc. In the CC term, the DEGs were concentrated in collagen-containing, extracellular matrix, fibrillar collagen trimer, collagen trimer, etc. In the MF term, the DEGs were concentrated in extracellular matrix structural constituent, cytokine activity, endopeptidase activity, metalloendopeptidase activity, serine-type endopeptidase activity, and others (Figure 3). Complete GO enrichment results are available in Table S2.

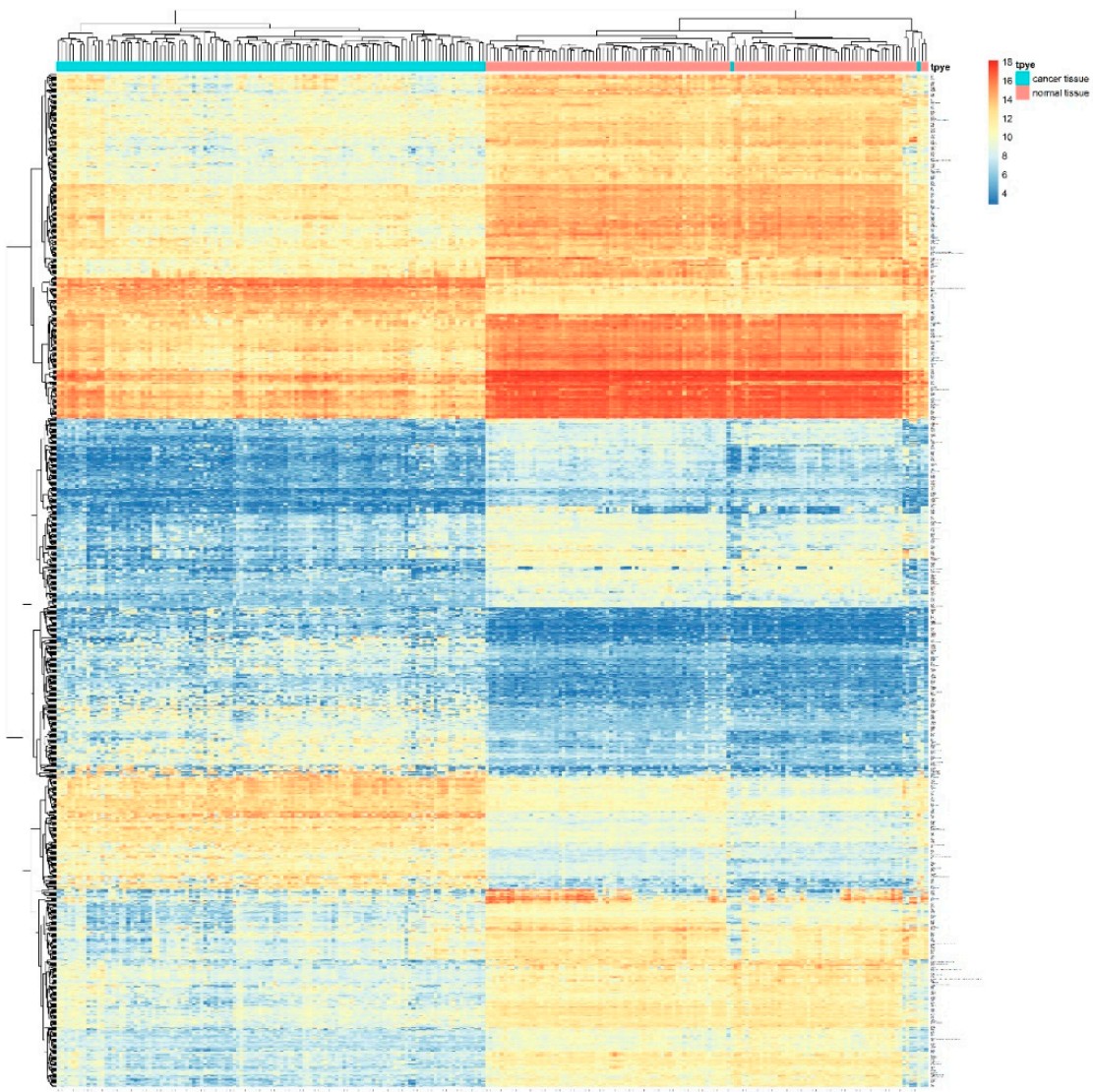

**Figure 2.** Heatmap of differential gene expression in each cancer tissue (blue band at the top) and normal tissue (red band at the top). Each square represents each gene, and each colored square in the heatmap represents the expression level of the gene. The higher the expression level, the darker the color will be (red represents highly expressed, blue means low expression value). Each row represents the expression level of each gene in different samples, and each column represents the expression level of all genes in each sample.

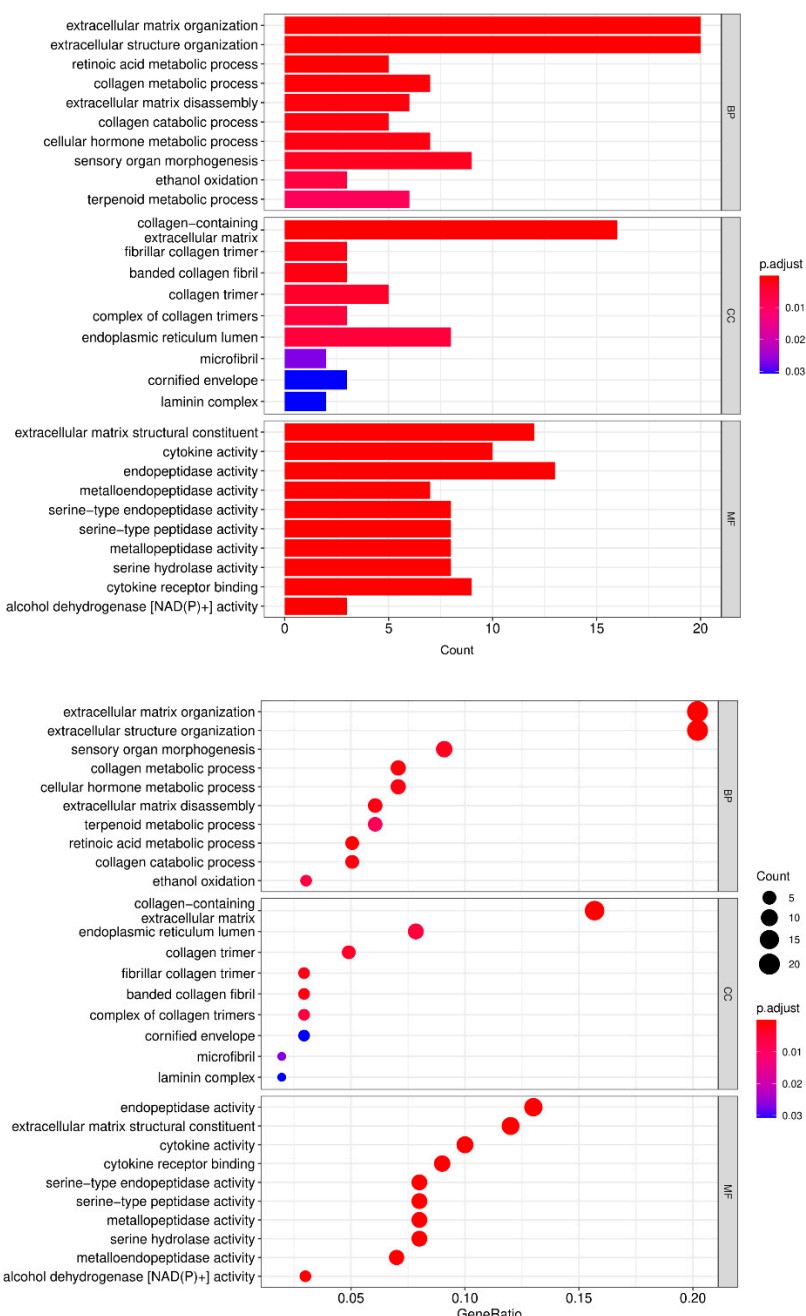

**Figure 3.** Gene ontology categories. BP: biological process, CC: cellular component, MF: molecular function.

### 3.4. Kyoto Encyclopedia of Genes and Genome (KEGG) Pathway Enrichment Analysis

Moreover, KEGG pathway enrichment analysis showed that DEGs were enriched into protein digestion and absorption, retinol metabolism, amoebiasis, cytokine-cytokine receptor interaction, chemical carcinogenesis, ECM-receptor interaction, rheumatoid arthritis, IL-17 signaling pathway, and pancreatic secretion (Figure 4). The complete KEGG pathway enrichment results are available in Table S3.

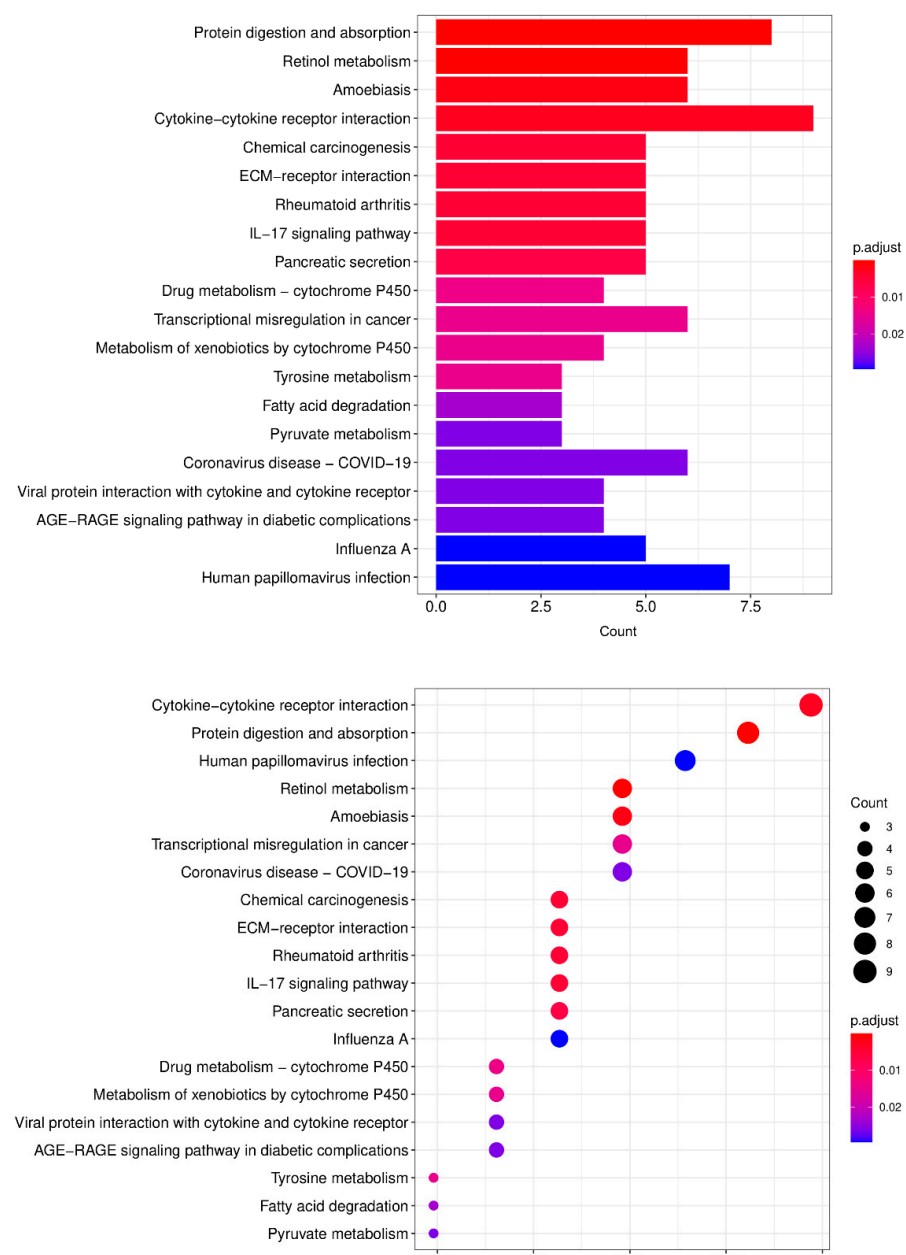

**Figure 4.** KEGG enrichment analysis results of the DEGs.

### 3.5. Protein–Protein Interaction Networks of DEGs

To clarify the functional proteins encoded by differential genes, a PPI network was constructed (Figure 5). Key proteins in the network can regulate many other proteins. We analyzed the above-mentioned PPI network with the cytoHubba plugin, sorted the proteins by the number of Unicom nodes, and obtained the top 10 key proteins, including ELN, MMP13, SPP1, MMP10, COL1A1, CSF2, MMP1, SERPINE1, CXCL8, and MMP3 (Figure 6). The complete degree results are available in Table S4.



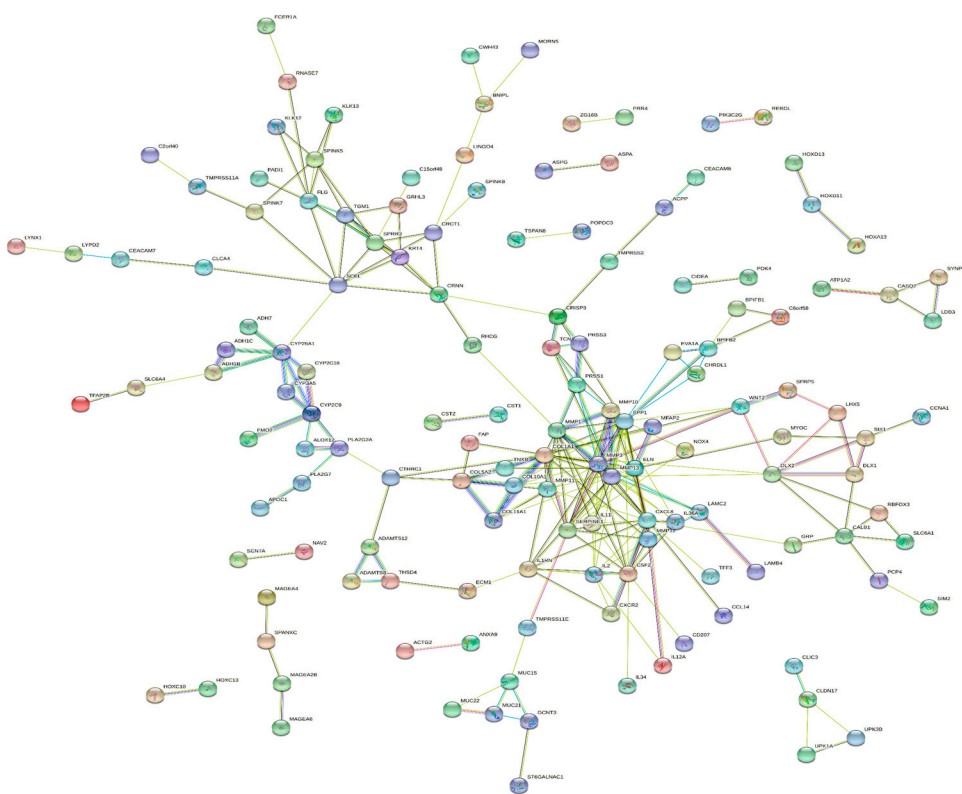

**Figure 5.** The PPI network analysis between different encoded proteins.

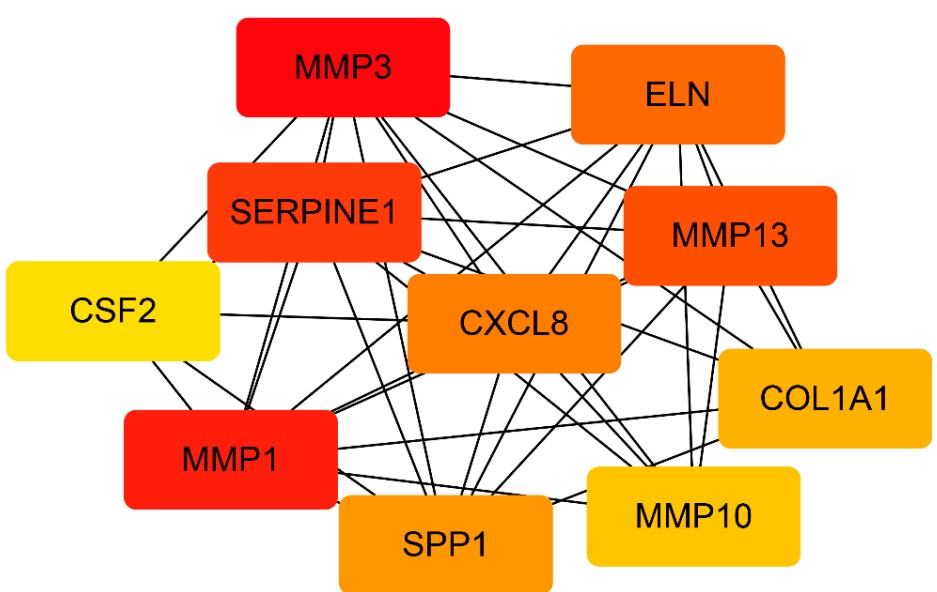

**Figure 6.** The top 10 hub genes in the PPI network of the upregulated DEGs and their interaction relationships (the darker red represents higher node degree).

### 3.6. Prognostic Analysis and Diagnostic Efficacy Analysis

The survival analysis shows that the higher expression of four key genes MMP13, SPP1, MMP10, and COL1A1 are associated with the prognosis of ESCC (Figure 7a). Besides, the ROC curve analysis indicated that the four genes had a high diagnostic efficacy (AUC: MMP13 0.885, SPP1 0.932 MMP10 0.909, COL1A1 0.890, Figure 7b).

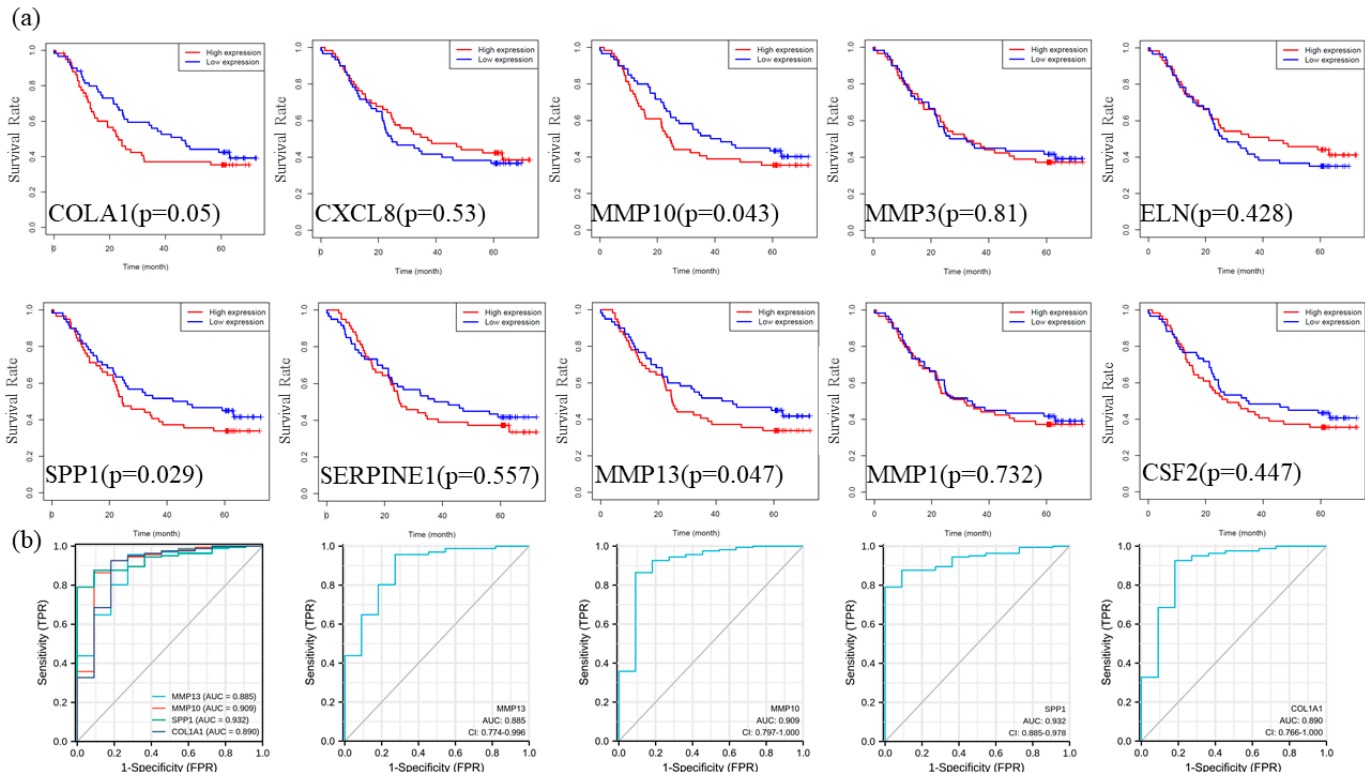

**Figure 7.** Survival analysis and ROC curve for overall survival in ESCC. (**a**) Survival analysis of ten differential expression genes (*p* < 0.05 for prognostic significance); (**b**) ROC analysis of the four genes that were significant in survival analysis (area > 0.9 under the curve is of high diagnostic value).

### 3.7. Validation of DEGs Expression

A GEPIA2 database and Western blot were performed to validate the expression level of DEGs. The GEPIA analysis results showed that the expression level of COL1A1, CXCL8, SPP1, MMP13, SERPNE1, MMP1, MMP3, and MMP10 was significantly upregulated in ESCA (Figure 8a, *p* < 0.05). In addition, the Western blot results showed that MMP13, SPP1, MMP10, and COL1A1 expression were upregulated in ESCC cell lines (KYSE-150 and KYSE-510) compared with that in the normal cell (Het-1A) (Figure 8b,c).

Consistent with these results, DEGs were found to be significantly over-expressed in ESCC by using distinct ESCC datasets via pooled analysis in the Oncomine database (Figure S1), and significant over-expression was also found in the TIMER database (Figure S2a–d).

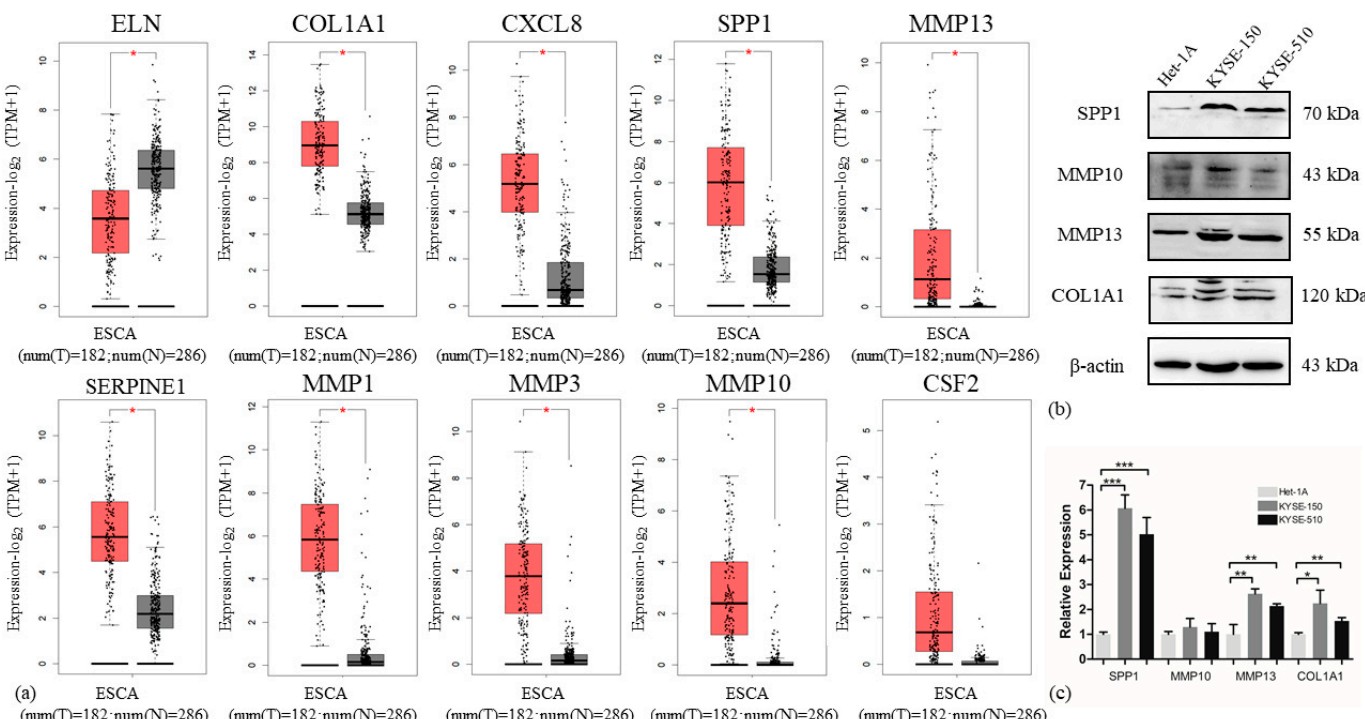

**Figure 8.** Validation of the DEGs by GEPIA2 and Western blot. (**a**) The top 10 genes expression in ESCA from GEPIA2 database (* *p* < 0.05); (**b**) Expression of four key genes (SPP1, MMP10, MMP13 and COL1A1) evaluated by Western blot; (**c**) Western blot quantification (each protein expression was normalized against expression level of β-actin and Het-1A cell line, * *p* < 0.05, ** *p* < 0.005, *** *p* < 0.0001).

## 4. Discussion

In this study, we re-annotated the lncRNA chip of GSE53624 in GEO, which contained 119 normal tissues and matched cancer tissues and completed the follow-up information. Then, we obtained the expression profile of 28,885 genes and a total of 210 down-regulated and 80 up-regulated DEGs were obtained by R package "limma". To reveal the underlying molecular mechanisms of the DEGs, functional enrichment analysis, pathway enrichment analysis, and PPI analysis were performed. The PPI network analysis showed that the 10 DEGs with top degrees (ELN, MMP13, SPP1, MMP10, COL1A1, CSF2, MMP1, SERPINE1, CXCL8, and MMP3) may play central roles in ESCC. GO analysis showed that DEGs were mainly related to the extracellular matrix, such as extracellular matrix organization, extracellular structural organization, extracellular matrix disassembly, and cytokine activity, which indicated that it is valuable to further study the DEGs and the key genes may become new diagnostic and therapeutic targets for ESCC. Pathway enrichment analysis revealed that DEGs enrichment of ESCC involved cytokine–cytokine receptor interaction, chemical carcinogenesis, ECM–receptor interaction, IL17 signaling pathway, and other related pathways. On the one hand, this finding confirmed that the DEGs play an essential role in ESCC's occurrence and development. On the other hand, it also suggested that the abnormal changes in the IL-17 signaling pathway might be of great significance in developing ESCC.

The IL-17 signaling pathway is involved in the body's immune response [20] and inflammatory response [21]. Hence, we used GEPIA2 to explore the relationship between DEGs expression and immune markers and found that the DEG expression level was significantly correlated with immune cell markers in ESCC. Specifically, MMP10 expression level was significantly correlated with 12 out of 42 immune cell markers, MMP13 expression level was significantly correlated with 30 out of 42 immune cell markers, SPP1 expression level was significantly correlated with 26 out of 42 immune cell markers, and COL1A1 ex-



pression level was significantly correlated with 27 out of 42 immune cell markers (Table S5). Our results indicated that the immune response and inflammatory response may promote the development of ESCC.

We further validated the expression of the DEGs through the online database GEPIA2 and Western blot. In the GEPIA2 database, COL1A1, CXCL8, SPP1, MMP13, SERPNE1, MMP1, MMP3, and MMP10 were significantly higher expressed. We also, based on the TCGA data, analyzed the expression levels of the MMP13, MMP10, COL1A1, and SPP1 in the main pathological stages of ESCA. Violin plots showed that stage II and stage III have higher gene expression levels than stage I, which suggested an association of increased key genes expression with tumor progression (Figure S3). The results of the Western blot further confirmed that the expression levels of SPP1, COL1A1, MMP13, and MMP10 were consistent with the expression trends in our bioinformatics analysis.

SPP1, also known as osteopontin (OPN), is an ECM protein that has been involved in a range of physiological and pathological processes [22]. Studies have shown that SPP1 is overexpressed in a variety of tumors and can be involved in tumor cell proliferation, angiogenesis, chemotherapeutic resistance, migration, and invasion [22–24]. It is valuable in tumor diagnosis and prognosis prediction of breast [23,24], lung [25], liver [26], ovarian [27], and bladder cancers [28]. In breast cancer, SPP1 activates the NF-κB signaling pathway [29]. In ESCC, SPP1 was associated with poor prognosis in patients with locally advanced ESCC receiving preoperative chemoradiotherapy [30].

Matrix metalloproteinases (MMPs) are widely regarded as important regulators of the tumor microenvironment [31]. According to the gene structure and substrate specificity of the enzyme, MMPs can be divided into six different subgroups. Collagenases (MMP1, 8, 13, 18) are able to degrade type I, II, and III collagen. Gelatinases (MMP2 and MMP9) cleave denatured type IV collagen. Matrix lysozymes (MMP3, 7, 10, 11, 26, 27) can effectively degrade type III, IV, V collagen, proteoglycan, glycoprotein, and gelatin, etc. MMP12 is also known as macrophage metalloelastase. MMP14, 15, 16, 17, 24, 25 are closely related to the activation of MMP2. Additionally, other unclassified MMPs (MMP19, 20, 21, 22, 23, 28). In the early stage ESCC tissues, MMP10 is highly expressed and is associated with poor prognosis in patients (Figure S3). MMP10 expression is also closely related to lymph node metastasis, TNM stage, and tumor invasion [32]. MMP13 is a proteolytic enzyme belonging to the endopeptidase family of extracellular matrix degradation and it is characterized by zinc binding motif at its catalytic site [33]. MMP13 is produced by many types of cancer and is a key regulator in the process of human malignant tumor metastasis [34–36]. It is overexpressed in a variety of malignancies [37]. MMP13 was originally identified from overexpressed breast cancer, and its role in breast tumorigenesis has been reported [38–40]. TP53 has a high mutation rate in esophageal cancer. Studies have shown that 59–93% of patients with esophageal cancer have a TP53 mutation [41,42], suggesting that TP53 may be involved in the regulation of MMP expression level in esophageal cancer. Previous studies have shown that Ajuba promotes the migration and invasion of esophageal squamous cell lines KYSE450 and KYSE510 by up-regulating the expression of MMP10 and MMP13 [43].

COL1A1, as a type of group I collagen, is abnormally highly expressed in breast cancer [44], gastric cancer [45], non-small cell lung cancer [46], and is associated with the progression and prognosis of cancer. It is an important protein component in the adhesion of the tumor cell and extracellular matrix [47]. In ESCC, Yin et al. found that COL1A1 enhanced cell proliferation, migration, and invasion [48].

Combined with PPI network analysis, survival analysis, and the ROC curve, the results showed that SPP1, MMP13, MMP10, and COL1A1 were four reliable predictors of diagnosis and prognosis of esophageal cancer. Meanwhile, we analyzed the MMP13, MMP10, SPP1, and COL1A1 expression profile in numerous human solid tumors via Oncomine and TIMER2 databases. The results demonstrated that MMP13, MMP10, SPP1, and COL1A1 gene expression were higher in breast cancer, colorectal cancer, head and neck cancer, liver cancer, stomach cancer, and others, than in their matched adjacent normal tissues. This opens up possibilities for follow-up pan-cancer analysis.

However, there are some limitations in our study. In public databases, datasets on ESCC are relatively scarce. More relevant clinical samples should be collected for ESCC research. The reliability of the analyses of molecular mechanism and immune infiltration levels, especially the correlation between immune signatures and these key genes, is not supported by experiments in vivo and in vitro. These key genes need to be verified in a larger cohort of ESCC. Some studies have pointed out that there is a correlation between ESCC and Human papillomavirus (HPV) [49,50]. However, others hold the opposite view [51]. A study suggested that the overexpression of p16 could be used to predict HPV infection [52]. We analyzed the correlation between p16 expression and the expression of MMP10, MMP13, SPP1, and COL1A1 in ESCC samples through the GEPIA2 database and showed that these genes are not correlated (Table S6). This result may be related to the study sample size.

## 5. Conclusions

Our study revealed some cell functional clusters and signaling pathways associated with ESCC by re-annotating a gene chip dataset with prognostic information and subsequent bioinformatics and medical statistical analysis. ELN, MMP13, SPP1, MMP10, COL1A1, CSF2, MMP1, SERPINE1, CXCL8, and MMP3 were identified as the key genes driving the occurrence and development of esophageal squamous carcinoma. By combining clinical prognosis information and Western blot analysis, we found four genes with diagnostic and prognostic significance, including MMP13, SPP1, MMP10, and COL1A1. Additionally, MMP13, SPP1, MMP10, and COL1A1 in ESCC also had significant correlations with immune markers. Follow-up studies on the mechanism and clinical value of these genes will vigorously promote the ESCC diagnosis and treatment.

**Supplementary Materials:** The following are available online at https://www.mdpi.com/article/10.3390/app11073229/s1, Figure S1: Expression of MMP13, MMP10, COL1A1 and SPP1 in different types of human cancers in the Oncomine database, Figure S2: Expression of DEGs in different types of human cancers in the TIMER database, Figure S3: Based on the TCGA data, the expression levels of the MMP13, MMP10, COL1A1 and SPP1 gene were analyzed by the main pathological stages (stage I, stage II, stage III and stage IV) of ESCA. Log2 (TPM + 1) was applied for log-scale. Violin plots showed that stage II and stage III had higher gene expression level than stage I, which suggested an association of increased key gene expression with tumor progression, Table S1: Clinical characteristics of ESCC patients in the research, Table S2: GO enrichment results, Table S3: The enriched pathways of DEGs, Table S4: The node scores of top 10 DEGs, Table S5: Correlation analysis between DEGs and immune cell infiltrations in ESCC samples using GEPIA2, Table S6: Correlation analysis between DEGs and HPV marker in ESCC samples using GEPIA2.

**Author Contributions:** Conceptualization, M.W.; Formal analysis, M.W.; Investigation, M.W. and L.M.; Methodology, M.W.; Project administration, Y.C. and L.M.; Software, M.W.; Validation, D.L., Y.H. and F.W.; Writing—original draft, M.W. and D.L.; Writing—review and editing, D.L., Y.H. and Z.J. All authors have read and agreed to the published version of the manuscript.

**Funding:** This research was funded by Shenzhen Strategic Emerging Industry Development Special Funds, grant number JCYJ20170816143646446, and Shenzhen Science and Technology Research and Development Funds, grant number JCYJ20200109143018683.

**Institutional Review Board Statement:** Not applicable.

**Informed Consent Statement:** Not applicable.

**Data Availability Statement:** Data is contained within the article.

**Acknowledgments:** We thank Linyun Hong, Yang Yan, Wenning Cao, Chaowen Xue, Xiang Ji and Mingzhe Wang of the Lab for assistance during the study. We thank Tsinghua University for its generous support of necessary experiment instruments.

**Conflicts of Interest:** The authors declare no conflict of interest.

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
