# Peer review of "Identification of Key Genes Related to the Prognosis of Esophageal Squamous Cell Carcinoma Based on Chip Re-Annotation"

_applsci, doi:10.3390/app11073229_

Round 1
Reviewer 1 Report
The article entitled " Identification of key genes related to the prognosis of esophageal squamous cell carcinoma based on chip re-annotation" focuses on the interesting aspect of modern diagnostics in oncology. I have a few comments.
- Figure 2 is not unreadable – it’s just a nice picture with no information for readers
- The discussion needs to be improved.Information on individual genes is scarce, especially in the context of esophageal cancer (e.g. COL1A1).
- The limitations of the study should take into account the fact that the researchers compared the database of samples from patients with esophageal cancer with the study on cell lines.This is a big limitation considering the lack of influence of many factors in the case of cell culture.
- There is lack of information about the difference in gene expression in patients with different clinical stages of the disease.
I recommend the article for publication after major revision.
Reviewer 2 Report
In the current study, authors identify prognostic and diagnostic biomarkers of ESCC in Chinese population by gene chip re-annotation analysis. By employing systematic bioinformatics analysis, the authors narrow down on MM13, SPP1, MMP10 and COL1A1 as key genes determining the outcome of ESCC in patients. Although the analysis is well designed, executed and written, however authors need to address the concerns on the significance of the current study.
Major comment:
- The informatics analysis did not implicate novel genes/mechanisms governing ESCC progression, as most of these findings are already published within the indicated population, this raises questions on the rationale behind the study.
- Authors need to discuss the relationship between HPV statuses with their findings.
- The western blot in Fig 8b needs to be quantified by densitometry
Minor comment:
- Provide legible images for Fig 5, S2
Round 2
Reviewer 1 Report
I have no more comments. I recommend the article for publication.
Author Response
Dear Reviewer and editors:
Thanks very much for your kind work and consideration on publication of our paper. On behalf of my co-authors, we would like to express our great appreciation to editors and reviewers.
Thank you and best regards.
Yours sincerely,
meiqi Wang